# Ultrasensitive quantification of serum IFN-α and IFN-γ in systemic lupus erythematosus: A cross-sectional observational study

Miguel Á. González-Gay[1,2‡*], Fuensanta Gómez-Bernal[3], Juan C. Quevedo-Abeledo[4], Cristina Almeida-Santiago[4], Elena Heras-Recuero[1], Arantxa Torres-Roselló[1], Antonia de Vera-González[3], Beatriz Tejera-Segura[5], Enrique García-Barrera[6], Teresa Blázquez-Sánchez[1], Luisa M. Villar[6], Javier Gonzalo Ocejo-Vinyals[7], Raquel Largo[1], Iván Ferraz-Amaro[8,9‡*]

1 Division of Rheumatology, IIS-Fundación Jiménez Díaz, Madrid, Spain, 2 Department of Medicine and Psychiatry, University of Cantabria, Santander, Spain, 3 Division of Central Laboratory, Hospital Universitario de Canarias, Tenerife, Spain, 4 Division of Rheumatology, Hospital Doctor Negrín, Las Palmas de Gran Canaria, Spain, 5 Division of Rheumatology, Hospital Insular, Las Palmas de Gran Canaria, Spain, 6 Department of Immunology, Hospital Ramón y Cajal, Madrid, Spain, 7 Department of Immunology, Hospital Universitario Marqués de Valdecilla, IDIVAL, Santander, Spain, 8 Department of Internal Medicine, University of La Laguna (ULL), Tenerife, Spain, 9 Division of Rheumatology, Hospital Universitario de Canarias, Tenerife, Spain

‡ These authors share senior authorship on this work.
* miguelaggay@hotmail.com (MÁG-G); iferrazamaro@hotmail.com (IF-A)

## Abstract

### Background

Interferon (IFN) has been implicated in the pathogenesis of patients with systemic lupus erythematosus (SLE). However, its measurement in serum has been limited by low circulating levels that fall below the detection threshold of standard laboratory assays. In this study, we measured serum levels of IFN-alpha (IFN-α) and IFN-gamma (IFN-γ) using a novel ultrasensitive assay. We then aimed to analyze the relationship between these IFN levels and a broad spectrum of disease characteristics, including indices of disease activity and remission, and autoantibodies profiles.

### Methods and findings

From an initial cohort of 400 patients, a total of 313 patients with SLE were recruited in this cross-sectional study from September 2023 to February 2024. A comprehensive characterization of the patients was performed, including autoantibody profiles and indices of disease activity (SLE-DAS, SLEDAI-2K, and LLDAS), damage (SLICC-DI), and remission (DORIS). IFN-α and IFN-γ serum levels were measured using Simoa (Single Molecule Array) technique. A multivariable linear regression analysis was performed to examine the associations between the disease characteristics and circulating IFN-α and IFN-γ as the dependent variables. Besides, the diagnostic capacity of serum IFN levels to discriminate between high and low disease

**Data availability statement:** All data underlying the study's findings is transparently provided in the Zenodo link https://zenodo.org/records/17106069.

**Funding:** This study was funded by Instituto de Salud Carlos III (ISCIII) through the project PI23/00046 and co-funded by the European Union (I.F.-A.) and by the Spanish Ministry of Health, Instituto de Salud Carlos III (ISCIII), PI24/00554, and co-funded by the European Union (M.Á.G.-G.); and the Spanish Red de Investigación RICORS - RD24/0007/0031 fondos de Next Generation EU, financing acting on "Mecanismo de Recuperación y Resiliencia (MRR)" (M.Á.G.-G.). The funders had no role in study design, data collection and analysis, decision to publish, or preparation of the manuscript. None of the authors received compensation, salary, or financial support from any of the funders associated with this study.

**Competing interests:** I have read the journal's policy and the authors of this manuscript have the following competing interests. The authors declare that there are no conflicts of interest. Nevertheless, I.F.-A. would like to acknowledge that he has received grants/research supports from Abbott, MSD, Janssen, and Roche, as well as consultation fees from company-sponsored peakers' bureaus associated with Abbott, Pfizer, Roche, Sanofi, Celgene, and MSD. Prof. M.A.G. has received research supports from GSK as well as consultation fees/participation from company-sponsored speakers' bureaus associated to GSK, Otsuka, Amgen and Sanofi.

**Abbreviations :** ACR, American College of Rheumatology; ANAs, antinuclear antibodies; AUC, area under the curve; DORIS, Definitions of Remission in SLE; ENA, extractable nuclear antigen; hs-CRP, high-sensitivity C-reactive protein; IFN, Interferon; IFN-α, IFN-alpha; IFN-γ, IFN-gamma; IQR, interquartile range; LLDAS, Lupus Low Disease Activity State; PGA, physician global assessment; ROC, receiver operating characteristic; SD, standard deviation; SLE, systemic lupus erythematosus; SLE-DAS, SLE Disease Activity Score; TLRs, Toll-like receptors.

activity was studied using area under the curve analysis and determination of optimal cutoff points. Serum levels of IFN-α and IFN-γ showed a significant, albeit weak, correlation (Pearson's $r = 0.369$, $p < 0.001$). Both IFNs exhibited minimal associations with demographic characteristics (such as age, sex, and body mass index) and traditional cardiovascular risk factors (including hypertension, diabetes, dyslipidemia, smoking status, obesity, and metabolic syndrome). After multivariable adjustment, IFN-α—but not IFN-γ—was significantly and positively associated with acute-phase reactants (C-reactive protein and interleukin-6), disease activity indices (SLEDAI-2K, beta coefficient: 0.20 [95% confidence interval 0.09, 0.32] *log* pg/ml, $p < 0.001$ and SLE-DAS, beta coefficient: 0.15 [95% confidence interval 0.05, 0.25] *log* pg/ml, $p = 0.003$) and the presence of antinuclear antibodies. In contrast, remission (as defined by DORIS) and low disease activity (LLDAS) were negatively and significantly associated with IFN-α levels after adjustment for covariates. However, when attempts were made to define IFN cutoff values to discriminate between active and inactive disease or remission, they exhibited a poor balance between sensitivity and specificity. The cross-sectional design of this study limits our ability to infer causality and raises the possibility of reverse causation.

## Conclusions

In this study, we observed that IFN-α, but not IFN-γ, significantly associates with inflammation, indices of disease activity and remission, and autoantibody status in SLE. Investigating the potential of IFN-α as a biomarker for treatment response and long-term outcomes is warranted.

## Author summary

### Why was this study done?

- Interferons are proteins involved in systemic lupus erythematosus (SLE), but are hard to measure accurately due to their low levels in blood.

- This study aimed to measure IFN-α and IFN-γ levels using a new sensitive test and understand their relationship with disease features in patients with SLE.

### What did the researchers do and find?

- We measured serum IFN-α and IFN-γ in 313 patients with SLE and analyzed links with disease activity, damage, and remission.

- IFN-α levels correlated with disease activity markers and autoantibody presence; IFN-γ showed no significant associations.

- Lower IFN-α levels were observed in patients in remission or with low disease activity.

- Attempts to establish cutoff values to distinguish active versus inactive disease had low accuracy.

**What do these findings mean?**

- IFN-α could be a useful marker for inflammation and disease activity in SLE.

- IFN-γ is less useful for tracking SLE activity.

- Challenges remain in defining precise cutoff levels for IFN-α in clinical practice.

- Study limitations include its cross-sectional design and difficulty in cutoff determination.

## Introduction

Systemic lupus erythematosus (SLE) is a chronic, multisystem autoimmune disease that has a wide array of disease manifestations, including constitutional symptoms, cutaneous manifestations, arthritis, hematologic abnormalities, and nephritis [1]. SLE is believed to arise from immune abnormalities influenced by the interplay of genetic, environmental, and hormonal risk factors [2]. An essential feature of SLE is the production of antinuclear antibodies (ANAs) that bind to DNA, proteins, and complexes of proteins with nucleic acids [3] to form immune complexes that can induce local inflammation following deposition in tissue [4]. These complexes stimulate cytokine production, particularly type I interferon (IFN), by activating cytoplasmic nucleic acid sensors in innate immune cells [5]. Besides, nucleic acid-containing immune complexes induce IFN-alpha (IFN-α) production primarily through Toll-like receptors (TLRs). These immune complexes, formed by nucleic acids bound to autoantibodies, are internalized by plasmacytoid dendritic cells and recognized by endosomal TLRs. This recognition triggers signaling cascades of transcription factors which results in robust production of IFN-α [6].

The type I IFN-α main function is to act as a critical component of the antiviral defense system. It provides a paracrine alarm signal that induces an antiviral state in nearby uninfected cells, increasing their ability to detect viral nucleic acids [7]. The type II IFNs are limited to IFN-gamma (IFN-γ). IFN-γ activates multiple effector functions in macrophages, including antigen presentation via class I and class II major histocompatibility complex molecules, and promotes Th1 differentiation in CD4 T cells, among many other effects [8]. It is known that IFNs modulate expression of up to 10% of human genes with consequences including blockade of viral entry, replication, and survival [9].

Most patients with SLE have elevated circulating levels of IFN-α and increased expression of IFN-α-inducible RNA transcripts by peripheral blood cells, especially in the setting of a disease flare [10]. These elevations are attributable in part to predisposing genetic factors influencing IFN expression [11], but are primarily driven by the stimulation of IFN production through immune complexes formed between ANAs and nucleic acid-containing antigens [12]. Recognition of these elevations led to the development of targeted treatments for SLE such as anifrolumab, which is an antibody that blocks a type 1 IFN receptor [13]. Similarly, although less studied, some studies have also indicated that the IFN-γ gene signature may occur early in SLE [14,15] and may have an important role in lupus nephritis [16].

The determination of serum IFN levels has traditionally been challenging due to their low physiological concentrations, often in the femtogram per milliliter range. Conventional methods, such as ELISA, lack the sensitivity required to reliably detect these low levels. Consequently, highly sensitive techniques like Simoa (Single Molecule Array) have been recently developed, significantly enhancing the capability to quantify IFN in serum or plasma at attomolar concentrations, corresponding to a 5,000-fold increase in sensitivity compared to classic ELISAs. Due to technical limitations, few studies with well-characterized patient cohorts have comprehensively evaluated the relationship between serum IFN levels and disease features in SLE. In this study, we aimed to investigate the associations between serum levels of type I IFN (IFN-α) and type II IFN (IFN-γ) with clinical and immunological characteristics in patients with SLE. To achieve this, we performed an extensive phenotypic characterization of a large SLE cohort and examined how the serum levels of both IFNs correlated with detailed measures of disease activity, organ damage, and remission status.

## Materials and methods

### Study design

This was a cross-sectional study that included 313 patients with SLE. Patient recruitment occurred from September 2023 to February 2024. The study was conducted at four tertiary hospitals in Spain: Hospital Universitario de Canarias (Approval Number CHUC-2023-48), Hospital Universitario Doctor Negrín (Approval Number 2021-023-1), Hospital Insular de Gran Canaria (Approval Number 2021-023-1), and Hospital Fundación Jiménez Díaz (Approval Number PIC135-23). The study protocol received approval from the Institutional Review Committees of all participating hospitals, and all participants provided written informed consent. Research was carried out in accordance with the Declaration of Helsinki. This study is reported as per the STROBE (Strengthening the Reporting of Observational Studies in Epidemiology) guideline (S1 Checklist). The analyses reported were pre-specified as part of the study design and detailed in the study protocol approved by the Institutional Review Board. Deviations from the initial plan were limited to the exclusion of ROC analyses for IFN-γ, which were not pursued since this cytokine did not show relevant associations with clinical or laboratory measures.

To address potential sources of bias, several measures were implemented in the study design and analysis. First, patients were consecutively recruited from multiple tertiary hospitals to enhance representativeness and reduce selection bias. Standardized protocols were used for data collection, including clinical assessments and laboratory measurements, to minimize measurement bias. All analyses were performed exclusively at the Hospital Universitario de Canarias to ensure consistency and reduce inter-laboratory variability, thereby maintaining biomarker stability and ensuring reliable IFN quantification. The use of multivariable regression analyses was planned to allow adjustment for potential confounders identified through a change-in-estimate approach, controlling for demographic and clinical-related variables that could influence IFN levels or disease characteristics.

### Study participants

All patients with SLE were 18 years or older, had a clinical diagnosis of SLE, and met ≥4 American College of Rheumatology (ACR) classification criteria for SLE [17]. They had been diagnosed by rheumatologists and were regularly followed up in rheumatology outpatient clinics. Participation was allowed for patients taking prednisone, at an equivalent dose ≤10 mg/day, as glucocorticoids are often used in the treatment of SLE. Exclusion criteria comprised treatment with anifrolumab or other investigational biological therapies targeting IFN pathways, inability or refusal to provide written informed consent, insufficient or improperly stored serum samples for IFN measurement, and the presence of overlapping systemic autoimmune diseases that could confound the study results. A total of 400 patients with a clinical diagnosis of SLE were initially preselected for this study. The patient selection process is detailed in S1 Fig. Patients were systematically excluded based on the following criteria: (1) failure to meet ≥4 ACR classification criteria for SLE (*n* = 45, 11.3%); (2) incomplete informed consent documentation (*n* = 20, 5.0%); (3) current treatment with anifrolumab (*n* = 12, 3.0%); and (4) insufficient or poorly stored serum samples unsuitable for cytokine analysis (*n* = 10, 2.5%). After applying these exclusion criteria sequentially, a total of 313 patients (78.2% of initially screened patients) were included in the final analysis (S1 Fig). Individuals included in the study completed a cardiovascular risk factor and medication use questionnaire and underwent a physical examination. Weight, height, body mass index, abdominal circumference, and systolic and diastolic blood pressure (measured with the participant in a supine position) were assessed under standardized conditions. Information regarding smoking status and hypertension treatment was obtained from the questionnaire. Medical records were reviewed to ascertain specific diagnoses and medications.

### Data collection

SLE disease activity and damage were assessed using the Systemic Lupus Erythematosus Disease Activity Index -2000 (SLEDAI-2K) [18] and the Systemic Lupus International Collaborating Clinics/American College of Rheumatology

(SLICC/ACR Damage Index -SDI-) [19], respectively. For the present study proposal, the SLEDAI-2k index was divided into none (0 points), mild (1–5 points), moderate (6–10 points), high (11–19), and very high activity (>20) as previously described [20]. SLE Disease Activity Score (SLE-DAS) was also calculated [21] and categorized as follows: remission (SLE-DAS ≤ 2.08), mild activity (2.08 < SLE-DAS ≤ 7.64), and moderate to severe activity (SLE-DAS > 7.64), according to established definitions [22]. Definitions of Remission in SLE (DORIS) was based on the absence of clinical disease activity as measured by the clinical SLEDAI-2K = 0 and physician global assessment (PGA) <0.5. The patient may be receiving antimalarials, low-dose glucocorticoids (e.g., prednisone ≤ 5 mg/day), and/or maintenance doses of immunosuppressive therapies [23]. Similarly, Lupus Low Disease Activity State (LLDAS) accepts a SLEDAI-2K ≤ 4 with no activity from major organ systems, no new clinical activity compared with the previous assessment, a PGA of ≤1, prednisone dose ≤7.5 mg/day and maintenance doses of antimalarials and immunosuppressive therapies [24]. Modified LLDAS was also calculated using ≤5 mg/day prednisone criteria instead of ≤7.5 mg [25].

## Laboratory measurements

High-sensitivity C-reactive protein (hs-CRP) levels were measured using a high-sensitivity immunoassay. Serum samples were collected following standard protocols, immediately aliquoted, and frozen at −80°C without delay. Samples were stored continuously at −80°C until assay to preserve the labile nature of IFN. These procedures conform with recommended best practices to maintain the stability and reliability of IFN measurements. IFN-α was measured from serum samples using the Simoa (Single Molecule Array) IFN-α assay (IFN-αMS) Advantage PLUS KitTM (no. 103638, Quanterix, Billerica, MA, USA). IFN-γ and interleukin (IL) 6 were measured using the Cytokine 4-Plex C Advantage PLUS Reagent KitTM (no. 105066, Quanterix, Billerica, MA, USA) following the manufacturer's instructions. Sample processing and analysis were done using an HD-X analyzer (software version 4.16.2307.14001; Quanterix).

## Statistical analysis

Demographic and clinical characteristics in patients with SLE and controls were described as mean ± standard deviation (SD) or percentages for categorical variables. For non-normally distributed continuous variables, data were expressed as median and interquartile range (IQR). The sample size was calculated to detect a clinically meaningful correlation between IFN-α levels and SLEDAI scores. Based on previous studies reporting correlations between inflammatory biomarkers and disease activity in SLE, we aimed to detect a minimum correlation of $r = 0.20$ with 80% statistical power and a significance level of $α = 0.05$. Using the standard formula for Pearson correlation with Fisher's Z transformation, the required sample size was calculated as: $n = [(Z_{α/2} + Z_{β})/Z_r]^2 + 3$ where $Z_r = 0.5 × \ln[(1 + r)/(1 − r)]$ represents Fisher's transformation of the expected correlation coefficient. For a correlation of $r = 0.20$, the minimum calculated sample size was $n = 194$ participants. Accounting for an anticipated 20% dropout rate, we established a recruitment target of $n = 232$ participants. Missing data were addressed by carefully reviewing all collected variables for completeness prior to analysis. Cases with missing key clinical or laboratory data were excluded from specific analyses to avoid introducing bias. The extent of missing data was minimal, and details of missing data are transparently reported. The relationship of disease characteristics with circulating IFN-α and IFN-γ was evaluated by multivariable linear regression analysis. Potential confounders were identified using a change-in-estimate approach. Specifically, variables were considered confounders and retained in the multivariable model if their inclusion resulted in a change of 10% or more in the beta coefficient of the main exposure variable. This criterion was applied iteratively, adjusting the model to control for confounding while maintaining parsimony. The initially selected potential confounders included age, sex, body mass index, abdominal circumference, smoking status, diabetes, hypertension, metabolic syndrome, and the use of statins and aspirin. Log-transformed variables were used for correlation and regression analyses to address non-normal distributions. Normality of the transformed variables was assessed using appropriate statistical tests for normality, such as the Shapiro-Wilk test, complemented by graphical methods including Q–Q plots and histograms to ensure the adequacy of the transformations. To evaluate the diagnostic ability of IFNs levels

according to different disease activity scores, corresponding receiver operating characteristic (ROC) curves were analyzed, and optimal cutoff points for each score were determined. Since the cost ratio of false negatives to false positives could not be established, it was set to 1. The prevalence used corresponded to that observed for each score in the study sample. Accordingly, the optimal cutoff point was determined using the method of maximum efficiency, which balances false negative and false positive costs (FNcost = FPcost), and it was adjusted considering the prevalence observed in our study population. Sensitivity, specificity, as well as positive and negative predictive values, were subsequently calculated for the optimal cutoff values. The area under the curve (AUC) was calculated for each ROC analysis, with 95% confidence intervals estimated using the DeLong method to assess the statistical significance of the discriminative capacity. All the analyses used a 5% two-sided significance level and were performed using Stata software, version 17/SE (StataCorp, College Station, TX, USA). *P*-values <0.05 were considered statistically significant. Data visualizations were performed using Julius AI (Julius AI, San Francisco, CA, USA; https://julius.ai), utilizing Python 3.x.

## Results

### Demographics and disease-related data of systemic lupus erythematosus patients

The median (IQR) serum levels of IFN-α and IFN-γ in patients with SLE were respectively 189 (38–1440) and 462 (263–966) femtog/ml. Table 1 provides an overview of the characteristics of the 313 patients included in the study. Most of the participants were women (91%), with a mean age ± SD of 52 ± 12 years. The average body mass index was 28 ± 6 kg/m$^2$, and the abdominal circumference was 90 ± 15 cm. Classic cardiovascular risk factors were observed, including current smoking in 18% of patients, hypertension in 38%, and obesity in 28%. Additionally, 30% of patients were on statin therapy, and 24% were taking aspirin (Table 1).

The average disease duration was 19 ± 11 years. Forty-five percent of the patients with SLE had no disease activity and mild or moderate-high activity was present in, respectively, 46% and 9% of the patients as indicated by the SLEDAI score. The median SLEDAS index value was 1.1 (IQR 0.4–2.1). Regarding the definitions of remission and low disease activity, according to DORIS and LLDAS criteria, 68% and 80% of patients, respectively, met these criteria. The SLICC-DI was 0 (IQR 0-1) and an SLICC-DI score of 1 or higher was found in 46% of patients. At recruitment, 76% of patients tested positive for anti-DNA antibodies and 71% for extractable nuclear antigen (ENA) antibodies, with anti-SSA being the most frequently detected (34%). At the time of the study, 73% of patients were using hydroxychloroquine. Other less frequently used disease-modifying antirheumatic drugs included methotrexate (12%) and mycophenolate mofetil (13%). Additionally, one-third of the patients (32%) were taking prednisone, with a median daily dose of 5 mg/day (IQR: 2.5–5). Further SLE-related clinical data are presented in Table 1.

Fig 1 presents a histogram of IFN-α and IFN-γ values (in picograms) in patients with SLE. As observed, these values exhibited a clearly non-normal, right-skewed distribution. Furthermore, Fig 1 illustrates the correlation between both types of IFNs. Following logarithmic transformation of IFN-α and IFN-γ values, their correlation was found to be statistically significant but moderate (Pearson's *r* = 0.369, *p* < 0.001).

### Associations of disease-related data with interferon-α and interferon-γ serum levels

The association between disease characteristics and serum levels of IFN-α and IFN-γ, after multivariable adjustment, is presented in Table 2. Regarding demographic characteristics, IFN-α showed a significant, negative association with age and obesity. In contrast, IFN-γ showed no significant association with demographic characteristics or cardiovascular risk factors.

With respect to disease-related characteristics, multiple significant associations were observed with IFN-α (Table 2). In this regard, after multivariable adjustment, both CRP and IL-6 levels showed significant positive correlations with IFN-α. Similarly, after adjusting for covariates, disease activity scores analyzed as continuous variables were significantly associated with IFN-α levels. However, when SLEDAI and SLEDAS scores were considered categorical, moderate-to-high

**Table 1. Characteristics of systemic lupus erythematosus patients.**

| | Patients with SLE | Missing data |
|---|---|---|
| | (*n* = 313) | *n* (%) |
| Interferon-α, femtog/ml | 189 (38–1,440) | 12 (3.8) |
| Interferon-γ, femtog/ml | 462 (263–966) | 6 (1.9) |
| Age, years | 52 ± 12 | 0 (0) |
| Sex, Female, n (%) | 285 (91) | 0 (0) |
| Body mass index, kg/m$^2$ | 28 ± 6 | 4 (1.3) |
| Abdominal circumference, cm | 90 ± 15 | 0 (0) |
| Waist circumference, cm | 101 ± 15 | 0 (0) |
| Waist to hip ratio | 0.91 ± 0.28 | 0 (0) |
| **Cardiovascular co-morbidity** | | |
| Smoking, *n* (%) | 56 (18) | 0 (0) |
| Diabetes, *n* (%) | 15 (5) | 0 (0) |
| Hypertension, *n* (%) | 120 (38) | 0 (0) |
| Dyslipidemia, *n* (%) | 176 (58) | 0 (0) |
| Obesity, *n* (%) | 88 (28) | 4 (1.3) |
| Metabolic syndrome, *n* (%) | 138 (45) | 0 (0) |
| Statins, *n* (%) | 93 (30) | 0 (0) |
| Aspirin, *n* (%) | 75 (24) | 0 (0) |
| **SLE-related data** | | |
| Disease duration, years | 19 ± 11 | 2 (0.6) |
| CRP, mg/dl | 1.7 (0.8–4.1) | 9 (2.9) |
| IL-6, pg/ml | 4.0 (2.4–7.4) | 2 (0.6) |
| SLICC-DI | 0 (0–1) | 0 (0) |
| SLICC-DI ≥ 1, *n* (%) | 145 (46) | 0 (0) |
| SLEDAI-2K | 2 (0–4) | 0 (0) |
| No activity, *n* (%) | 141 (45) | |
| Mild, *n* (%) | 145 (46) | |
| Moderate to high, *n* (%) | 27 (9) | |
| Clinical SLEDAI-2K | 0 (0–1) | 0 (0) |
| Clinical SLEDAI-2K > 0 | 96 (31) | 0 (0) |
| SLEDAS | 1.1 (0.4–2.1) | 4 (1.3) |
| Remission | 237 (78) | |
| Mild activity | 58 (19) | |
| Moderate or severe activity | 14 (5) | |
| DORIS, *n* (%) | 212 (68) | 2 (0.6) |
| LLDAS, *n* (%) | 249 (80) | 1 (0.3) |
| Modified LLDAS, *n* (%) | 248 (79) | 1 (0.3) |
| Auto-antibody profile, n (%) | | |
| Anti-DNA | 238 (76) | 0 (0) |
| Anti-ENA | 220 (71) | 3 (1.0) |
| Anti-SSA | 107 (34) | 1 (0.3) |
| Anti-SSB | 44 (14) | 1 (0.3) |
| Anti-RNP | 83 (27) | 1 (0.3) |
| Anti-Sm | 50 (16) | 1 (0.3) |
| Anti-ribosome | 35 (11) | 1 (0.3) |
| Anti-nucleosome | 62 (20) | 1 (0.3) |

*(Continued)*

| | Patients with SLE | Missing data |
|---|---|---|
| Anti-histone | 51 (16) | 1 (0.3) |
| Antiphospholipid syndrome, *n* (%) | 50 (16) | 0 (0) |
| Antiphospholipid autoantibodies, *n* (%) | 104 (33) | 0 (0) |
| Lupus anticoagulant, *n* (%) | 68 (22) | 9 (2.9) |
| Anticardiolipin IgM, *n* (%) | 40 (13) | 5 (1.6) |
| Anticardiolipin IgG, *n* (%) | 48 (16) | 5 (1.6) |
| Anti beta2 glycoprotein I IgM, *n* (%) | 28 (9) | 6 (1.9) |
| Anti beta2 glycoprotein I IgG, *n* (%) | 33 (11) | 6 (1.9) |
| Current prednisone, *n* (%) | 100 (32) | 0 (0) |
| Prednisone, mg/day | 5 (2.5–5) | 0 (0) |
| Hydroxychloroquine, *n* (%) | 229 (73) | 0 (0) |
| Methotrexate, *n* (%) | 39 (12) | 0 (0) |
| Mycophenolate mofetil, *n* (%) | 41 (13) | 0 (0) |
| Azathioprine, *n* (%) | 26 (8) | 0 (0) |
| Rituximab, *n* (%) | 10 (3) | 0 (0) |
| Belimumab, *n* (%) | 39 (12) | 0 (0) |

Data represent means ± SD or median (interquartile range) when data were not normally distributed. CRP, C reactive protein; ANA, antinuclear antibodies; ENA, extractible nuclear antibodies; SLEDAI, Systemic Lupus Erythematosus Disease Activity Index; SLEDAI categories were defined as: 0, no activity; 1–5 mild; 6–10 moderate; >10 high activity, >20 very high activity; SLICC, Systemic Lupus International Collaborating Clinics/American Colleague of Rheumatology Damage Index; SLE-DAS, SLE Disease Activity Score categorized as: remission ≤2.08, mild activity ≤7.64, and moderate/severe activity >7.64; DORIS, Definitions of Remission in SLE; LLDAS, Lupus Low Disease Activity State; Modified LLDAS uses ≤5 mg/day prednisone criteria instead of ≤7.5 mg; Clinical SLEDAI-2k omits complement and anti-dsDNA components from original SLEDAI-2K.

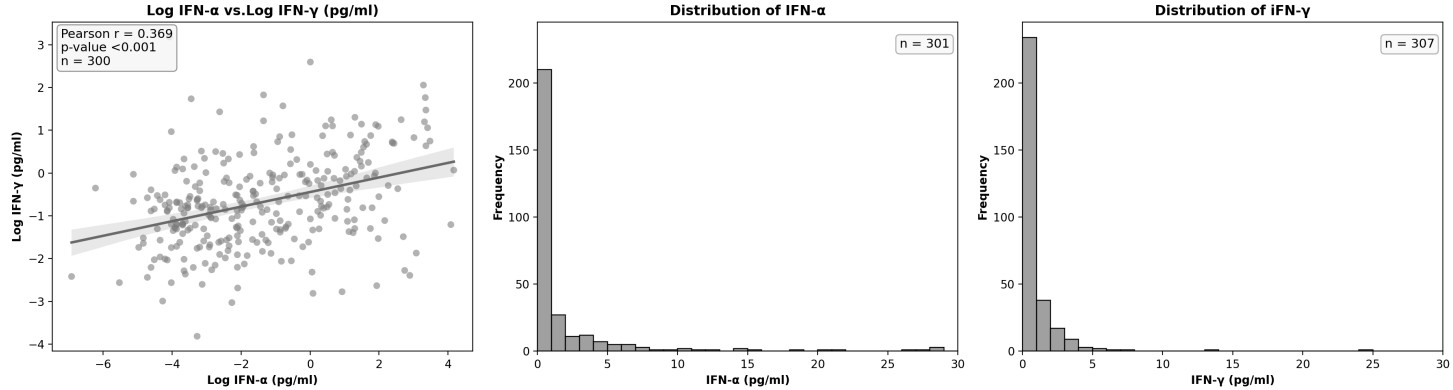

**Fig 1. Scatter plot of the relationship between log IFN-α and log IFN-γ and frequency histograms of non-transformed IFN-α and IFN-γ values.** IFN, Interferon.

SLEDAI activity was significantly associated with elevated IFN-α compared to no activity, but this was not the case for mild activity compared to no activity. Besides, for SLE-DAS, mild activity showed a multivariable significant association with higher IFN-α levels relative to remission, whereas moderate or severe activity did not. In contrast, remission according to DORIS criteria and low disease activity as defined by LLDAS showed significant and independent negative associations with IFN-α (Fig 2).

# Table 2. Associations of disease relate data with interferon-α and-γ serum levels.

| | Beta coef. [95% CI], p | | | | | | | |
| --- | --- | --- | --- | --- | --- | --- | --- | --- |
| | *log* IFN-α, pg/ml | | | | *log* IFN-γ, pg/ml | | | |
| | Univariable | | Multivariable | | Univariable | | Multivariable | |
| Age, years | **−0.04 [−0.06,−0.02]** | **<0.001** | | | −0.01 [0.02, 0.003] | 0.142 | | |
| Sex, Female | 0.36 [−0.53, 0.26] | 0.425 | | | 0.38 [−0.03, 0.78] | 0.070 | | |
| Body mass index, kg/m$^2$ | −0.01 [−0.06, 0.03] | 0.570 | | | −0.001 [−0.02, 0.02] | 0.902 | | |
| Abdominal circumference, cm | −0.01 [−0.03, 0.01] | 0.311 | | | −0.001 [−0.01, 0.01] | 0.752 | | |
| Waist circumference, cm | −0.01 [−0.02, 0.01] | 0.545 | | | −0.0001 [−0.01, 0.01] | 0.982 | | |
| Waist to hip ratio | −0.62 [−1.52, 0.29] | 0.179 | | | −0.22 [−0.64, 0.19] | 0.289 | | |
| Cardiovascular co-morbidity | | | | | | | | |
| Smoking | −0.00 [−0.67, 0.67] | 0.997 | | | −0.18 [−0.49, 0.13] | 0.250 | | |
| Diabetes | −1.00 [−2.17, 0.18] | 0.097 | | | −0.31 [−0.86, 0.24] | 0.265 | | |
| Hypertension | −0.33 [−0.86, 0.19] | 0.215 | | | 0.08 [−0.16, 0.32] | 0.509 | | |
| Obesity | **−0.58 [−1.15, −0.01]** | **0.048** | | | −0.13 [−0.39, 0.13] | 0.338 | | |
| Metabolic syndrome | −0.08 [−0.60, 0.44] | 0.767 | | | 0.05 [−0.18, 0.29] | 0.658 | | |
| Statins | −0.29 [−0.85, 0.27] | 0.304 | | | 0.04 [−0.22, 0.30] | 0.753 | | |
| Aspirin | 0.34 [−0.27, 0.94] | 0.275 | | | −0.14 [−0.42, 0.14] | 0.319 | | |
| SLE-related data | | | | | | | | |
| Disease duration, years | 0.002 [−0.02, 0.02] | 0.833 | | | 0.01 [−0.005, 0.02] | 0.277 | | |
| CRP, mg/dl | **0.06 [0.00, 0.12]** | **0.049** | **0.07 [−0.01, 0.1]** | **0.018** | 0.03 [−0.0001, 0.05] | 0.051 | **0.03 [0.001, 0.05]** | **0.042** |
| IL6, pg/ml | 0.02 [−0.004, 0.05] | 0.089 | **0.03 [0.001, 0.05]** | **0.042** | **0.01 [0.0008, 0.03]** | **0.036** | **0.01 [0.002, 0.03]** | **0.025** |
| SLEDAI-2K | **0.24 [0.13, 0.35]** | **<0.001** | **0.20 [0.09, 0.32]** | **<0.001** | −0.004 [−0.06, 0.05] | 0.884 | | |
| SLEDAI-2K categories | | | | | | | | |
| No activity | ref. | | ref. | | ref. | | | |
| Mild | **0.42 [−0.10, 0.95]** | 0.117 | 0.3 [−0.2, 0.8] | 0.278 | −0.12 [−0.37, 0.12] | 0.326 | | |
| Moderate to high | **1.57 [0.54, 0.59]** | **0.002** | **1.32 [0.34, 0.82]** | **0.008** | −0.01 [−0.46, 0.43] | 0.952 | | |
| Clinical SLEDAI-2K | **0.26 [0.11, 0.42]** | **0.001** | **0.23 [0.08, 0.38]** | **0.003** | −0.02 [−0.09, 0.05] | 0.603 | | |
| Clinical SLEDAI-2K > 0 | **0.73 [0.17, 1.28]** | **0.010** | **0.60 [0.05, 1.15]** | **0.032** | −0.08 [−0.33, 0.18] | 0.556 | | |
| SLEDAS | **0.17 [0.07, 0.27]** | **0.001** | **0.15 [0.05, 0.25]** | **0.003** | −0.01 [−0.06, 0.04] | 0.704 | | |
| Remission | ref. | | ref. | | ref. | | ref. | |
| Mild activity | **0.95 [0.30, 1.61]** | **0.004** | **0.75 [0.10, 1.41]** | **0.024** | 0.14 [−0.16, 0.45] | 0.354 | 0.10 [−0.21, 0.41] | 0.510 |
| Moderate or severe activity | 1.13 [−0.12, 2.38] | 0.075 | 1.08 [−0.14, 2.31] | 0.083 | −0.44 [−1.05, 0.17] | 0.160 | −0.45 [−1.06, 0.16] | 0.150 |
| DORIS | **−0.73 [−1.28, −0.18]** | **0.009** | **−0.67 [−1.21, −0.13]** | **0.014** | 0.06 [−0.19, 0.31] | 0.645 | | |
| LLDAS | **−1.32 [−1.96, −0.69]** | **<0.001** | **−1.20 [−1.82, −0.57]** | **<0.001** | 0.18 [−0.12, 0.47] | 0.240 | | |
| Modified LLDAS | **−1.36 [−1.99, −0.74]** | **<0.001** | **−1.24 [−1.86, −0.62]** | **<0.001** | 0.19 [−0.10, 0.48] | 0.205 | | |
| Auto-antibody profile | | | | | | | | |
| Anti-DNA | 0.53 [−0.07, 0.12] | 0.83 | 0.42 [−0.16, 1.01] | 0.154 | 0.07 [−0.20, 0.35] | 0.594 | | |
| Anti-ENA | **1.33 [0.78, 0.87]** | **<0.001** | **1.27 [0.74, 1.81]** | **<0.001** | 0.04 [−0.21, 0.30] | 0.739 | | |
| Anti-SSA | **1.28 [0.75, 0.80]** | **<0.001** | **1.32 [0.81, 1.83]** | **<0.001** | 0.25 [−0.00, 0.50] | 0.051 | 0.24 [−0.008, 0.49] | 0.058 |
| Anti-SSB | 0.70 [−0.05, 0.45] | 0.066 | **0.88 [0.15, 1.61]** | **0.018** | 0.07 [−0.27, 0.42] | 0.674 | | |
| Anti-RNP | **1.49 [0.05, 0.93]** | **<0.001** | **1.43 [087, 1.98]** | **<0.001** | −0.05 [−0.32, 0.22] | 0.726 | | |
| Anti-Sm | **1.34 [0.03, 0.65]** | **<0.001** | **1.18 [0.50, 1.86]** | **0.001** | 0.02 [−0.31, 0.34] | 0.923 | | |
| Anti-ribosome | **1.59 [0.42, 0.76]** | **<0.001** | **1.35 [0.52, 2.18]** | **0.001** | 0.02 [−0.36, 0.40] | 0.926 | | |
| Anti-nucleosome | 0.65 [−0.00, 0.29] | 0.051 | 0.47 [−0.18, 1.11] | 0.155 | −0.15 [−0.44, 0.15] | 0.337 | | |
| Anti-histone | **0.85 [0.16, 0.54]** | **0.017** | **0.75 [0.07, 1.43]** | **0.030** | −0.01 [−0.33, 0.31] | 0.964 | | |
| Antiphospholipid syndrome | −0.19 [−0.90, 0.51] | 0.585 | | | −0.05 [−0.37, 0.28] | 0.780 | | |
| Antiphospholipid autoantibodies | −0.28 [−0.83, 0.26] | 0.309 | | | −0.07 [−0.32, 0.18] | 0.593 | | |

*(Continued)*

**Table 2.** (Continued)

| | Beta coef. [95% CI], p | | | | | | | |
| --- | --- | --- | --- | --- | --- | --- | --- | --- |
| | *log* IFN-α, pg/ml | | | | *log* IFN-γ, pg/ml | | | |
| | Univariable | | Multivariable | | Univariable | | Multivariable | |
| Lupus anticoagulant | −0.06 [−0.68-0.56] | 0.843 | | | 0.003 [−0.28, 0.29] | 0.985 | | |
| Anticardiolipin IgM | **−0.90 [−1.66, −0.15]** | **0.019** | **−0.90 [−1.65, −0.15]** | **0.019** | −0.14 [−0.49, 0.21] | 0.433 | | |
| Anticardiolipin IgG | **−0.81 [−1.52, −0.11]** | **0.024** | **−0.80 [−1.50, −0.09]** | **0.027** | −0.07 [−0.39, 0.26] | 0.679 | | |
| Anti beta2 glycoprotein I IgM | −0.82 [−1.71, 0.06] | 0.068 | **−1.03 [−1.92, −0.15]** | **0.023** | 0.07 [−0.34, 0.48] | 0.741 | | |
| Anti beta2 glycoprotein I IgG | 0.03 [−0.82, 0.88] | 0.944 | | | 0.19 [−0.19, 0.57] | 0.332 | | |
| Current prednisone | **0.73 [0.18, 0.28]** | **0.009** | **0.77 [0.23, 1.31]** | **0.005** | 0.05 [−0.21, 0.30] | 0.725 | | |
| Prednisone, mg/day | 0.02 [−0.10, 0.15] | 0.735 | | | −0.06 [−0.13, 0.01] | 0.112 | −0.06 [−0.14, 0.01] | 0.089 |
| Hydroxychloroquine | −0.36 [−0.94, 0.22] | 0.225 | | | −0.05 [−0.32, 0.21] | 0.701 | | |
| Methotrexate | 0.38 [−0.39, 0.15] | 0.335 | | | 0.11 [−0.24, 0.46] | 0.541 | | |
| Mycophenolate mofetil | −0.00 [−0.77, 0.76] | 0.990 | | | 0.05 [−0.30, 0.40] | 0.789 | | |
| Azathioprine | **1.20 [0.13, 0.28]** | **0.011** | **0.93 [0.006, 1.85]** | **0.049** | 0.25 [−0.18, 0.68] | 0.259 | | |
| Rituximab | 0.80 [−0.63, 0.23] | 0.273 | | | −0.24 [−0.94, 0.46] | 0.498 | | |
| Belimumab | **0.96 [0.20, 0.73]** | **0.014** | **0.81 [0.05, 1.56]** | **0.037** | 0.26 [−0.09, 0.61] | 0.146 | 0.23 [−0.13, 0.58] | 0.209 |

In this analysis, interferon serum levels are the dependent variable. Analyses were adjusted for age, sex, body mass index, abdominal circumference, smoking status, diabetes, hypertension, metabolic syndrome, and use of statins and aspirin, using a criterion of a 10% or greater change in the beta coefficient. CRP, C reactive protein; ANA, antinuclear antibodies; ENA, extractable nuclear antibodies; SLEDAI, Systemic Lupus Erythematosus Disease Activity Index; SLEDAI categories were defined as: 0, no activity; 1–5 mild; 6–10 moderate; >10 high activity, >20 very high activity; SLE-DAS, SLE Disease Activity Score categorized as: remission ≤2.08, mild activity ≤7.64, and moderate/severe activity >7.64. DORIS, Definitions of Remission in SLE; LLDAS, Lupus Low Disease Activity State; Modified LLDAS uses ≤5 mg/day prednisone criteria instead of ≤7.5 mg; Clinical SLEDAI-2k omits complement and anti-dsDNA components from original SLEDAI-2K. Significant *p* values are depicted in bold.

The autoantibody profile demonstrated widespread positive associations with IFN-α after multivariable adjustment, including the presence of any anti-ENA antibodies as well as anti-SSA, anti-SSB, anti-RNP, anti-Sm, anti-histone, and anti-ribosomal antibodies. Notably, only anti-DNA and anti-nucleosome antibodies did not show significant associations. Additionally, the presence of anticardiolipin IgM and anti-β2 glycoprotein I IgM antibodies was also associated with IFN-α, although in these cases the association was negative. With respect to therapeutic variables, the use of prednisone as a binary variable and belimumab were both associated with higher IFN-α levels (Table 2).

Unlike IFN-α, the assessment of IFN-γ levels showed minimal association with disease characteristics. In this regard, only CRP and IL-6 levels exhibited significant associations after multivariable adjustment. Notably, disease activity and remission scores did not show significant associations with serum IFN-γ levels (Table 2).

### Diagnostic performance of IFN-α levels to distinguish disease activity as defined by several activity scores

The diagnostic ability of IFN-α for different disease states is shown in S1 Table. This analysis was not performed for IFN-γ since its values were not associated with disease features. For this analysis, the SLEDAI-2K and SLE-DAS scores were dichotomized so that the former represented a value equal or higher to 4 *versus* below 4, or a value higher than 0, and the latter represented mild to severe activity *versus* remission (i.e., >2.08). Similarly, for DORIS and LLDAS, to avoid inverted ROC curves, the reference category was defined as remission or low disease activity. In all cases, IFN-α demonstrated significant AUC values (with 95% confidence intervals excluding 0.5). However, these AUC values were consistently weak, indicating limited discriminatory capacity (S1 Table and S2 Fig). IFN-α levels below approximately 11,000–16,000 femtograms/ml were associated with remission or low disease activity, while levels above 11,000–21,000 femtograms/ml discriminated high disease activity. The cutoff values for remission according to DORIS and LLDAS were 11,400 and 15,500

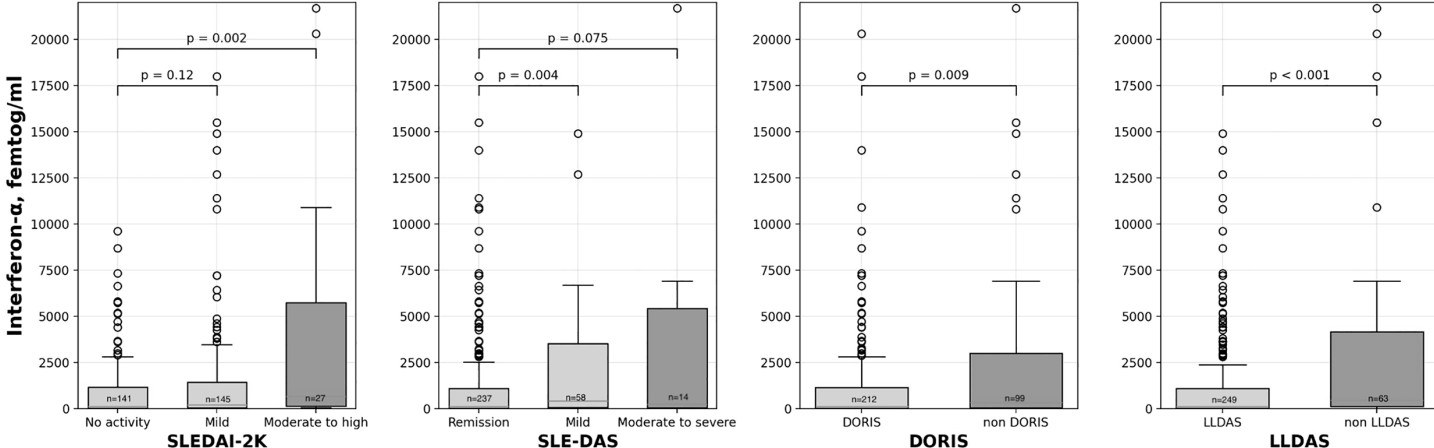

**Fig 2. Boxplots showing interferon-α levels (femtog/ml) across different lupus activity measures.** SLEDAI-2k index is divided into none (0 points), mild (1–5 points), moderate (6–10 points), high (11–19), and very high disease activity (>20). SLE Disease Activity Score (SLE-DAS) is categorized as follows: remission (≤2.08), mild activity (2.08 ≤ 7.64), and moderate to severe disease activity (>7.64). DORIS remission status: DORIS vs. non-DORIS. DORIS definition: absence of clinical disease activity as measured by the clinical SLEDAI-2K = 0 and physician global assessment <0.5, the patient may be receiving antimalarials, prednisone ≤5 mg/day, and/or maintenance doses of immunosuppressive therapies. LLDAS remission status: LLDAS vs. non-LLDAS. LLDAS definition: a SLEDAI-2K ≤4 with no activity from major organ systems, no new clinical activity compared with the previous assessment, a PGA of ≤1, prednisone dose ≤7.5 mg/day and maintenance doses of antimalarials and immunosuppressive therapies. *P* values shown above brackets represent univariable statistical comparisons of log-transformed interferon-α values, though the displayed data are presented in original (non-log-transformed) scale. The Y-axis is truncated at 22,000 femtog/ml for visualization purposes. Box-and-whisker plot illustrating individual data points, the median (central line), interquartile range (box), whiskers representing the minimum and maximum values excluding outliers, and outliers shown as points beyond the whiskers. SLEDAI-2K, Systemic Lupus Erythematosus Disease Activity Index; SLE-DAS, SLE Disease Activity Score; DORIS, Definitions of Remission in SLE; LLDAS, Lupus Low Disease Activity State.

femtograms/ml, respectively. SLEDAI-2K ≥4 corresponded to IFN-α levels >10,900 femtograms/ml and SLE-DAS >2.08 to IFN-α levels >21,700 femtograms/ml. Contrary, when SLEDAI-2K >0 was applied, the cutoff point was established in 39 femtograms/ml. Notably, all of these cutoffs showed a poor balance between sensitivity and specificity. While specificity values were consistently high, sensitivity was notably low (S1 Table and S2 Fig).

## Discussion

The present study is a large, well-characterized multicenter cohort, with a robust sample size and detailed clinical and laboratory profiling, to simultaneously assess serum levels of both IFN-α and IFN-γ in patients with SLE. We found that IFN-α—but not IFN-γ—demonstrated a strong association with key disease features, including, laboratory findings and disease activity scores. The use of a novel, ultrasensitive assay enabled precise quantification of IFN levels. These findings highlight the potential value of IFN-α as a biomarker of disease activity in SLE, given its consistent association with a broad spectrum of clinical and laboratory parameters.

To date, only a few studies have been described on the use of ultrasensitive IFN-α assays in patients with SLE. In this regard, a previous study measured serum IFN-α2 levels in 48 children with SLE and 67 healthy controls using a Simoa-based IFN-α assay [26]. A clear positive correlation was present between serum IFN-α2 levels and the IFN-I gene signature. Serum IFN-α2 levels and gene signature showed a significant negative trend in the first 3 years after diagnosis following treatment. In the linear mixed model, serum IFN-α2 levels were significantly associated with SLEDAI, while the IFN gene signature did not show this association. Besides, both IFN-I assays were able to characterize LLDAS and disease flares [26]. In a separate study, a total of 407 patients with SLE were recruited, including 254 in remission and 153 not in remission [27]. The authors defined elevated IFN-α as a threshold of 136 fg/mL, corresponding to three SDs above

the mean serum IFN-α concentration calculated from 68 healthy blood donors. They found that a significant proportion of patients with SLE in remission exhibited elevated serum IFN-α levels, particularly in the presence of anti-dsDNA and anti-ribonucleoprotein antibodies (e.g., anti-Ro/SSA 60, anti-RNP). In this report, elevated circulating IFN-α was identified as an independent predictive biomarker for disease flare within the following year. The authors suggested that incorporating serum IFN-α measurements into routine laboratory assessments for patients in remission could help clinicians identify individuals who, despite clinical remission, continue to overexpress IFN-α and are at increased risk of relapse [27].

IFN-α concentrations in serum samples from 150 consecutive patients with SLE were measured using digital ELISA in a cross-sectional study [28]. This study also found a correlation between IFN-α levels and disease activity as assessed by SLEDAI. However, other indices of activity, remission, or damage were not assessed in that report. Our study adopts a descriptive approach, and the large number of patients included, combined with their detailed characterization, allowed for comprehensive multivariable adjustment. Notably, despite its cross-sectional design, we identified associations between serum IFN-α levels and a wide range of disease features. In this regard, to the best of our knowledge, our study represents the largest investigation to date of IFN-α expression in a cohort of patients with SLE. According to our findings, IFN-α serves as a reliable indicator of multiple characteristics related to both disease activity and immunological expression.

Two former studies found no significant correlation between IFN-γ levels and disease activity [29,30]. However, to our knowledge, no previous reports have assessed circulating IFN-γ using ultrasensitive techniques in large cohorts of patients with SLE. Although certain genetic features related to IFN-γ have been described in SLE, particularly during the early and active stages of the disease [31], supporting a pathogenic role for this cytokine, our results indicate that serum IFN-γ levels do not correlate with disease activity or immunological features in SLE. Therefore, IFN-γ cannot be considered a useful biomarker for this population.

Recent studies have demonstrated increased type I IFN gene expression in patients with primary antiphospholipid syndrome, which correlates with the presence of anti-beta2 glycoprotein I antibodies [32]. However, in our study, we observed a negative association between IFN-α levels and the presence of certain anticardiolipin and anti-β2 glycoprotein I antibodies of the IgM isotype. This was also the case in a recent study of 76 women in which IFN-α serum levels were identified as a potential risk factor for lower birth weight in their infants [33]. In that study, IFN-α levels were also negatively correlated with anti-β2 glycoprotein I and anticardiolipin antibodies. One possible explanation is that patients with these autoantibodies could represent distinct serological or clinical subgroups, each characterized by unique genetic, immunological, and cellular activation profiles, which could differentially influence IFN expression. Another possible explanation for this unexpected negative correlation is that IFN-α production and the generation of specific IgM autoantibodies could be governed by distinct immunological pathways or vary according to disease stage. For example, IFN-α levels could fluctuate depending on disease activity or treatment effects, whereas IgM autoantibody levels could indicate a more stable or earlier immune response. Furthermore, immune regulatory mechanisms, such as tolerance or feedback inhibition, could suppress IFN-α production in patients with elevated levels of these autoantibodies. Overall, these findings point to a complex interplay between innate immune signaling and humoral autoimmunity.

In our study, the distribution of serum IFN-α and IFN-γ levels deviated from normality and exhibited a right-skewed pattern, suggesting a potential genetic influence on the levels of these IFNs in individuals. Furthermore, a positive correlation was observed between the two IFNs, likely reflecting the known stimulatory effect they can have on each other [34]. However, this correlation was weak, reinforcing our finding that only IFN-α, and not IFN-γ, showed a significant association with SLE characteristics. This implies that despite some degree of interaction between these cytokines, IFN-α might play a more direct or dominant role in the pathogenesis and clinical manifestations of SLE.

In recent years, the assessment of IFN activity in SLE has focused on measuring IFN signatures through RNA-based detection of IFN-stimulated genes. These techniques, which require genetic material and specialized equipment, are costly and limited in their clinical applicability. Besides, there is no international consensus on which set of genes

constitutes the optimal IFN signature, and different studies use varying gene panels [35]. Moreover, the relationship between the IFN gene signature and serum protein levels of different IFN subtypes remains unclear, with frequent discordance between gene expression and detectable serum protein [36]. The technical challenges and low sensitivity of traditional assays for quantifying serum IFN proteins have driven the adoption of gene expression signatures as surrogate markers of IFN pathway activation in SLE. Remarkably, a recent report demonstrates that IFN-α measured by Simoa and the IFN-I gene score perform equally well in identifying the association of type I IFN with SLE disease activity and clinical manifestations [37]. This association was specific to IFN-α, as no correlation was observed with serum IFN-γ levels. The authors suggest that serum IFN-α levels may be useful for assessing disease activity, as measuring IFN-α by digital ELISA may be easier to standardize than gene expression scores. Our findings are consistent with this direction. Based on our results, serum IFN-α levels are directly associated with disease activity, suggesting that circulating IFN-α could serve as a reliable biomarker for the disease.

In our study, when determining the optimal cutoff points for IFN-α, we found that these cutoffs exhibited low sensitivity and modest AUC values, indicating limited diagnostic discrimination. The identification of statistically optimal cutoff points is influenced by the cost ratio of false negatives to false positives, as well as by the prevalence of the outcome in the studied population. In our analysis, the cost ratio was set to 1, and the prevalence was based on our own study cohort. This limitation may have contributed to the observed low diagnostic performance of IFN-α levels. Additionally, disease activity scores in SLE have been widely criticized for their limited ability to accurately differentiate between varying degrees of disease activity [38]. Moreover, the absence of a universally accepted "gold standard" for assessing disease activity in SLE presents significant challenges for the validation of biomarkers using disease activity scores. In this regard, in a previous study of 165 patients with SLE, the threshold IFN-α value associated with active disease (SLEDAI score of >0) was determined to be 266 femtog/mL [28]. Additionally, a cutoff of 225.9 fg/ml for disease flares has been defined in another work [26]. However, these two cited studies represent populations that differ from ours. For example, in the first, the median SLEDAI was 4 (range 0–36) with 55% of patients having a SLEDAI above 4, and in the second, the median clinical SLEDAI was 3 (range 0–14). This contrasts markedly with our cohort, where the median SLEDAI was 2 (IQR 0–4), the median clinical SLEDAI was 0 (IQR 0–1), and only 26% of patients had SLEDAI ≥4. Besides, these two previous reports did not use more contemporary activity scores, nor did they specify the prevalence or the statistical method used to establish this cutoff.

Although we were unable to establish optimal cutoff thresholds, the association between IFN-α and disease activity was strong in our study. We believe the limited discriminatory capacity of IFN-α reflects more the limitations of disease activity assessment tools than the biomarker itself. Further research in diverse populations is needed to validate these findings and determine more accurate cutoff values. Despite suboptimal cutoff performance, IFN-α remains a valuable serum biomarker of SLE activity when interpreted alongside clinical and laboratory data.

Historically, measuring IFNs in serum or plasma has been challenging, as their low circulating levels often fall below the detection limits of conventional assays. In recent years, the introduction of Simoa technology has largely overcome this limitation. Although Simoa provides high specificity, excellent reproducibility, and a strong correlation with biological activity, its high cost and limited availability restrict its use to specialized laboratories. Our work has the strength of using this technology that is more accurate and reliable than that employed in earlier reports. Besides, this study evaluated serum IFN-α and IFN-γ concurrently in a large cohort of patients recruited for this purpose. Our sample size also enabled multivariable analyses, which were lacking in some previous studies. We believe the study population is reasonably representative of adult patients with SLE seen in routine rheumatology clinical practice. In this regard, the sample included patients diagnosed by rheumatologists who met established classification criteria. Patients were recruited from multiple hospitals in Spain, enhancing generalizability across different clinical settings within the country. Overall, the cohort's demographic and clinical features, including age, sex distribution, disease duration, and activity levels, align well with previously published SLE cohorts, supporting its representativeness to adult SLE populations. Although the high proportion of patients

in remission might be seen as a limitation, it also represents a strength, as it suggests that IFN-α levels are associated with disease characteristics even in clinically inactive patients. Furthermore, organ involvement was not assessed in our study, and therefore we cannot draw conclusions regarding the relationship between IFN values and organ involvement. Additionally, we acknowledge the limitation of our cross-sectional study design, which precludes the inference of causality and includes the possibility of reverse causation. We also acknowledge that our study did not include a control group. However, our primary objective was not to compare serum IFN levels between patients with SLE and healthy controls, but rather to investigate their association with disease characteristics within the SLE population. Lastly, a potential limitation is that the confounder selection process, although primarily informed by theory and prior knowledge, involved data-driven methods that do not differentiate confounders from mediators or colliders. As a result, there is a risk of overadjustment or collider bias, which could influence our findings.

In conclusion, elevated serum levels of IFN-α are associated with higher disease activity and increased ANA expression in patients with SLE. Additionally, IFN-α levels differentiate patients in remission or with low disease activity. Contrary, serum IFN-γ levels do not demonstrate diagnostic utility in SLE. Further research is needed to precisely establish threshold values that can effectively distinguish between these disease activity states. Nonetheless, IFN-α holds promise as a valuable and reliable biomarker for the assessment of disease activity in the routine clinical management of patients with SLE.

## Supporting information

**S1 Fig.** Patient selection flowchart showing inclusion of 313 patients from 400 initially screened individuals with systemic lupus erythematosus.
(TIF)

**S2 Fig. Receiver Operating Characteristic (ROC) area under the curves (AUC) for IFN-α serum levels and activity, damage, and remission scores.** SLEDAI, Systemic Lupus Erythematosus Disease Activity Index; SLE-DAS, SLE Disease Activity Score; DORIS, Definitions of Remission in SLE; LLDAS, Lupus Low Disease Activity State; Clinical SLEDAI-2k omits complement and anti-dsDNA components from original SLEDAI-2K.
(TIF)

**S1 Table. Diagnostic performance (area under the curve) of IFN-α levels to distinguish disease activity as defined by several activity scores.**
(DOCX)

**S1 Checklist. STROBE (Strengthening the Reporting of Observational Studies in Epidemiology) guideline checklist** . For more information, please see: https://www.strobe-statement.org/ - STROBE.
(DOCX)

**S1 Protocol. Complete study protocol detailing inclusion criteria, procedures, data collection methods, and statistical analysis.**
(DOCX)

**S1 Text. Abbreviations.**
(DOCX)

## Author contributions

**Conceptualization:** Miguel Á. González-Gay, Iván Ferraz-Amaro.

**Data curation:** Fuensanta Gómez-Bernal, Juan C. Quevedo-Abeledo, Cristina Almeida-Santiago, Elena Heras-Recuero, Arantxa Torres-Roselló, Antonia de Vera-González, Beatriz Tejera-Segura, Enrique García-Barrera, Teresa Blázquez-Sánchez, Luisa M. Villar, Javier Gonzalo Ocejo-Vinyals, Raquel Largo.

**Formal analysis:** Fuensanta Gómez-Bernal, Iván Ferraz-Amaro.

**Funding acquisition:** Iván Ferraz-Amaro.

**Investigation:** Iván Ferraz-Amaro.

**Methodology:** Iván Ferraz-Amaro.

**Project administration:** Iván Ferraz-Amaro.

**Resources:** Miguel Á. González-Gay.

**Supervision:** Iván Ferraz-Amaro.

**Validation:** Miguel Á. González-Gay, Iván Ferraz-Amaro.

**Visualization:** Miguel Á. González-Gay, Iván Ferraz-Amaro.

**Writing – original draft:** Miguel Á. González-Gay, Iván Ferraz-Amaro.

**Writing – review & editing:** Miguel Á. González-Gay, Iván Ferraz-Amaro.

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
