## [Editor Report · Decision Letter 0]

11 Aug 2025

Dear Dr Ferraz-Amaro,

Thank you for submitting your manuscript entitled "Ultrasensitive Quantification of Serum IFN-α, But Not IFN-γ, Reflects Inflammation, Disease Activity, and Autoantibody Status in Systemic Lupus Erythematosus" for consideration by PLOS Medicine.

Your manuscript has now been evaluated by the PLOS Medicine editorial staff and I am writing to let you know that we would like to send your submission out for external peer review.

For clinical studies, please upload a copy of your trial study protocol as a supporting information file. The study protocol should be the version submitted for approval to the institutional review board or ethics committee, should include any amendments to the study protocol, as well as the date of their approval by the institutional review or ethics committee. Please also detail any deviations from the study protocol in the Methods section of your manuscript. The editors will consider the protocol and study conduct prior to a final decision for external review.

Please re-submit your manuscript within two working days, i.e. by Aug 13 2025.

Feel free to email me at atosun@plos.org or us at plosmedicine@plos.org if you have any queries relating to your submission.

Kind regards,

Alexandra Tosun, PhD

Senior Editor

PLOS Medicine

---

## [Decision Letter · Decision Letter 1]

3 Sep 2025

Dear Dr Ferraz-Amaro,

Many thanks for submitting your manuscript "Ultrasensitive Quantification of Serum IFN-α, But Not IFN-γ, Reflects Inflammation, Disease Activity, and Autoantibody Status in Systemic Lupus Erythematosus" (PMEDICINE-D-25-02814R1) to PLOS Medicine. The paper has been reviewed by subject experts and a statistician; their comments are included below and can also be accessed here: https://www.editorialmanager.com/pmedicine/l.asp?i=1026346&l=PA4Y7ZOB

As you will see, the reviewers were positive about the manuscript but also offered valuable suggestions to strengthen it further. After discussing the paper with the editorial team, I'm pleased to invite you to revise the paper in response to the reviewers' and editors' comments. We plan to send the revised paper to some or all of the original reviewers, and we cannot provide any guarantees at this stage regarding publication.

We ask that you submit your revision by Sep 24 2025. However, if this deadline is not feasible, please contact me by email, and we can discuss a suitable alternative.

Don't hesitate to contact me directly with any questions (atosun@plos.org).

Best regards,

Alexandra

Alexandra Tosun, PhD

Senior Editor

PLOS Medicine

atosun@plos.org

Comments from the reviewers:

Reviewer #1: Gonzalez-Gay et al. present a very nice data set relating serum interferon-alpha and -gamma levels measured by high sensitivity Simoa assay to a number of demographic and clinical parameters in 313 patients with SLE. While the observations presented are not novel and generally confirm previously published studies, the availability of IFN levels across a large number of patients allows for a multivariate analysis of correlations that emphasize the significance of IFN-a (but not IFN-gamma) as a mediator with potential to assist in characterizing disease activity and possibly organ damage in the context of clinical practice. The effort to identify cutoff IFN values for degree of disease activity or remission are not helpful (could be eliminated from the manuscript) and the ROC curves shown in Supplementary Figure 1 indicate that in the absence of other measurements, quantitation of IFN-a levels will have limited utility as a biomarker in clinical practice. Nonetheless, the data do support IFN-a levels as highly relevant to lupus disease, and they would be useful to incorporate into clinical practice as one of many factors to be considered in patient assessment. As noted by the authors, the Simoa assay is currently rarely available to clinicians and even for investigators it is quite expensive. Perhaps the data presented here could encourage clinical testing facilities to make the Simoa assay available to clinicians.

In addition to the use of the Simoa assay, strengths of the study include the wide spectrum of patients studied, with more than half characterized as in DORIS remission, and the detailed autoantibody data provided. Notable is the very strong positive correlation between autoantibodies specific for RNA-associated particles and the significant negative correlation with certain anti-phospholipid (APL) antibodies. The latter observation has been previously reported, including in a recent publication from Karolinska (Rheumatology, 2025, 64, 1469-1475) describing IFN-a levels in pregnant SLE patients. The distinction between autoantibodies targeting RNA-associated proteins and those among APL specificities in relation to activation of the IFN pathway was noted years ago, and the current manuscript's data suggest an opportunity for further investigation of the role of the antiphospholipid antibodies in regulating IFN expression.

The manuscript might be strengthened by tightening up the Introduction and Discussion. Much of the Introduction simply reviews background related to interferons and is probably unnecessary for most readers. The authors (in the first paragraph) note the role of cytoplasmic nucleic acid sensors in induction of IFN-a but they fail to describe the role of nucleic acid-containing immune complexes in inducing IFN-a through Toll-like receptors, likely the major mechanism. The role of TLRs should be recognized.

The Discussion is also excessively long and does not provide any major insights into the biologic significance of the findings. Shortening the manuscript to just present the data with a brief acknowledgement of other relevant publications would be helpful.

Reviewer #2: I only reviewed the methods and statistics of this paper. There are several major concerns:

1. Confounders selected based on p-values are not appropriate for 2 reasons: p-values are not designed for variable selection and doing so could inflate type 1 errors; this practice does not account for the causality between variables - confounders by definition should be a variable causing exposure and outcome. If the author intend for a prediction (which has not been clear) they should consider either using AIC for variable selection or LASSO.

2. Relatedly, if that analysis aims for causal inference, it should be cautioned that cross-sectional design would mean reverse causation is possible.

3. Log-transformation was done to account for non-normally distributed variables but have the authors examine whether the transformed variables are 'sufficiently normal'?

4. It is clear how the 'optimal cut-off' is determined - is it by maximising the sum of sensitivity ana specificity?

5. The AUC and sensitivity shown are really low - I'm not sure whether this is clinically relevant / useful.

6. (minor) would be good to show Table 1 by the outcome.

Reviewer #3: I carefully read the article by Gonzalez-Gay et al. entitled "Ultra-sensitive quantification of serum IFN-α, but not IFN-γ, reflects inflammation, disease activity, and autoantibody status in systemic lupus erythematosus." This study demonstrates that serum IFN-α, measured with an ultra-sensitive immunoassay, is a valuable biomarker of SLE activity. It confirms findings from other groups in the field, using a large cohort of patients with this disease. The data also show that IFN-γ is a less reliable marker of SLE activity compared to IFN-α. As studies using ultra-sensitive immunoassays to quantify IFN-α in SLE remain scarce, this work is an important contribution to the scientific community. However, the clinical utility of ultra-sensitive assays appears less critical for IFN-γ than for IFN-α, given the higher circulating serum concentrations of the former. The manuscript is well-structured, concise, and easy to follow. However, several aspects require revision and, hopefully, improvement from the authors.

Major points

* For clinicians, perhaps the most striking—and disappointing—result is that serum IFN-α levels, although correlated with disease activity, do not reliably discriminate between clinically active and inactive patients, regardless of the activity scores used. This finding contrasts with previous reports (Mathian, Arthritis Rheumatol 2019, PMID: 30507062; Wahadat MJ, Rheumatology 2023, PMID: 36515466). It is surprising that the present study identifies discriminant thresholds as high as >10-15 pg/mL, whereas earlier studies reported much lower cut-offs (266 fg/mL in Mathian et al., 225.9 fg/mL in Wahadat et al.). Such high IFN-α levels are usually observed in clinically overtly active SLE and are rarely found in patients with minimal disease manifestations. These discrepancies likely reflect differences in assay technologies, pre-analytical processes, and possibly patient characteristics. Of note, the Gonzalez-Gay cohort included relatively few clinically active patients, with poorly detailed distribution of activity levels and no description of clinical involvement types. The authors acknowledge these limitations, including the inability to test their platform on healthy controls.

Furthermore, the use of SLEDAI and SLEDAS in this context raises concerns, as these scores include serological markers (complement and anti-dsDNA), which are not clinically meaningful indicators of clinical activity. For instance, the authors report 45% inactive versus 55% active patients. Yet, they also report 68% patients in remission. This implies that, at most, only 32% had clinical evidence of activity—likely fewer, since many of those not in remission were classified so merely due to prednisone >5 mg/day, despite clinical inactivity. It is probable that the 23% discrepancy (55%-32%=23%) represents patients with only serological abnormalities (complement consumption and/or elevated anti-dsDNA), which are of little clinical relevance.

To clarify these issues and assist readers in interpreting the data, I suggest the following:

o Provide detailed graphical presentations of individual IFN-α values and medians across subgroups (by SLEDAI activity level: none/mild/moderate/high; by SLE-DAS activity levels; remission/mild/moderate/severe; remission vs non-remission; LLDAS vs non-LLDAS). This would allow readers to assess dispersion and subgroup sizes, as well as to visualize how many patients fall above or below ROC-derived thresholds.

o For ROC analyses, repeat calculations excluding purely serological activity by using the clinical SLEDAI (i.e., omitting complement and anti-dsDNA components). Many clinically inactive patients are misclassified as active under standard SLEDAI ≥4, while some with overt manifestations fall below this cut-off. Similar considerations may apply to SLE-DAS if a "clinical" version exists. Additionally, for comparison, I strongly encourage ROC analyses of C3, C4, and anti-dsDNA against clinical SLEDAI, DORIS, and LLDAS.

o Expand the discussion by comparing the proportion of active patients in this study with other published cohorts.

* The discussion misrepresents the work of Mathian et al. (Mathian, Arthritis Rheumatol 2019, PMID: 30507062). The threshold of 266 fg/mL was derived using ROC curves with clinical activity (clinical SLEDAI >0) as the gold standard, not SLEDAI ≥4. This point requires correction.

* Damage indices such as SDI integrate cumulative disease and treatment consequences over many years (mean disease duration ~19 years in this cohort). It is biologically implausible that a short-term biomarker such as serum IFN-α, which can fluctuate within minutes, reflects cumulative damage. Without longitudinal IFN-α data, such analyses should be avoided.

* Additional analyses of modified LLDAS (≤5 mg/day prednisone instead of 7.5 mg) would be informative, given its status as a current treatment target (Fanouriakis A, Ann Rheum Dis 2024, PMID: 37827694).

* In Table 2, cytokine levels should be reported for each subgroup, including measures of dispersion, to fully inform the reader.

* Table 2 would be more clinically interpretable if results were expressed as odds ratios rather than beta coefficients.

Minor points

* Pre-analytical conditions must be detailed (sample type, storage, and handling), as IFN-α is labile and requires strict preservation protocols.

* In Table 1, several percentages appear inconsistent. My recalculations yield: dyslipidemia 56% (vs 58%), metabolic syndrome 44% (vs 45%), SLEDAS remission 76% (vs 78%), anti-ENA 70% (vs 71%), anticardiolipin IgG 15% (vs 16%). These discrepancies should be corrected.

* The claim that this is the first multivariable analysis of IFN-α clinical associations is inaccurate; such analyses have been reported (Mathian, Arthritis Rheumatol 2019, PMID: 30507062).

* The study does not reference Chasset et al. (Ann Rheum Dis 2022, PMID), one of the earliest to compare ultra-sensitive immunoassays for IFN-α, IFN-γ, and type I IFN scores in relation to SLE activity. This work should be discussed.

Any attachments provided with reviews can be seen via the following link: https://www.editorialmanager.com/pmedicine/l.asp?i=1026346&l=PA4Y7ZOB

---

* Please upload any figures associated with your paper as individual TIF or EPS files with 300dpi resolution at resubmission; please read our figure guidelines for more information on our requirements: http://journals.plos.org/plosmedicine/s/figures. While revising your submission, we strongly recommend that you use PLOS's NAAS tool (https://ngplosjournals.pagemajik.ai/artanalysis) to test your figure files. NAAS can convert your figure files to the TIFF file type and meet basic requirements (such as print size, resolution), or provide you with a report on issues that do not meet our requirements and that NAAS cannot fix.

After uploading your figures to PLOS's NAAS tool - https://ngplosjournals.pagemajik.ai/artanalysis, NAAS will process the files provided and display the results in the "Uploaded Files" section of the page as the processing is complete.

If the uploaded figures meet our requirements (or NAAS is able to fix the files to meet our requirements), the figure will be marked as "fixed" above. If NAAS is unable to fix the files, a red "failed" label will appear above.

When NAAS has confirmed that the figure files meet our requirements, please download the file via the download option, and include these NAAS processed figure files when submitting your revised manuscript.

* Thank you for agreeing to make your data available. At this time, please provide the link to the data repository and accession numbers required for access.

* The funding statement should include: specific grant numbers, initials of authors who received each award, URLs to sponsors’ websites. Also, please state whether any sponsors or funders (other than the named authors) played any role in study design, data collection and analysis, the decision to publish, or preparation of the manuscript. If they had no role in the research, include this sentence: “The funders had no role in study design, data collection and analysis, decision to publish, or preparation of the manuscript.”

FIGURES AND TABLES

SUPPLEMENTARY MATERIAL

REFERENCES

STUDY TYPE-SPECIFIC REQUESTS

* In the manuscript text, please indicate: (1) the specific hypotheses you intended to test, (2) the analytical methods by which you planned to test them, (3) the analyses you actually performed, and (4) when reported analyses differ from those that were planned, transparent explanations for differences that affect the reliability of the study's results. If a reported analysis was performed based on an interesting but unanticipated pattern in the data, please be clear that the analysis was data driven.

* Please state in the Methods section whether the study had a prospective protocol or analysis plan. If a prospective analysis plan (from your funding proposal, IRB or other ethics committee submission, study protocol, or other planning document written before analyzing the data) was used in designing the study, please include the relevant document(s) with your revised manuscript as a Supporting Information file to be published alongside your study and cite it in the Methods section. A legend for this file should be included at the end of your manuscript. If no such document exists, please make sure that the Methods section transparently describes when analyses were planned, and when/why any data-driven changes to analyses took place. Changes in the analysis, including those made in response to peer review comments, should be identified as such in the Methods section of the paper, with rationale.

* Please ensure that the study is reported according to the STARD guideline (https://www.equator-network.org/reporting-guidelines/stard/) and include the completed STARD checklist as Supporting Information. Please add the following statement, or similar, to the Methods: "This study is reported as per the Standards for Reporting of Diagnostic Accuracy (STARD) guideline (S1 Checklist)." When completing the checklist, please use section and paragraph numbers, rather than page numbers.

* Please structure your Abstract according to STARD for Abstracts (https://www.equator-network.org/reporting-guidelines/stard-abstracts/).

* Please structure the Methods section using the following sub-headings: Study design, Participants, Test methods, Analysis.

* Please include a diagram to describe the flow of participants through the study (typically figure 1).

* Address the extent to which the study population is representative of the population of interest.

---

## [Decision Letter · Decision Letter 2]

4 Nov 2025

Dear Dr. Ferraz-Amaro,

Thank you very much for re-submitting your manuscript "Ultrasensitive Quantification of Serum IFN-α, But Not IFN-γ, Reflects Inflammation, Disease Activity, and Autoantibody Status in Systemic Lupus Erythematosus" (PMEDICINE-D-25-02814R2) for review by PLOS Medicine.

Thank you for your detailed response to the reviewers' and editors’ comments. I have discussed the paper with my colleagues, and it has also been seen again by all three original reviewers. The changes made to the paper were mostly satisfactory to the reviewers. As such, we intend to accept the paper for publication, pending your attention to the reviewers' and editors' comments below in a further revision. When submitting your revised paper, please once again include a detailed point-by-point response to the editorial comments. The remaining issues that need to be addressed are listed at the end of this email.

In revising the manuscript for further consideration here, please ensure you address the specific points made by each reviewer and the editors. In your rebuttal letter you should indicate your response to the reviewers' and editors' comments and the changes you have made in the manuscript. Please submit a clean version of the paper as the main article file. A version with changes marked must also be uploaded as a marked up manuscript file. Please also check the guidelines for revised papers at http://journals.plos.org/plosmedicine/s/revising-your-manuscript for any that apply to your paper.

We ask that you submit your revision within 1 week (Nov 11 2025). However, if this deadline is not feasible, please contact me by email, and we can discuss a suitable alternative.

Please do not hesitate to contact me directly with any questions (atosun@plos.org).

We look forward to receiving the revised manuscript.

Sincerely,

Alexandra Tosun, PhD

Senior Editor 

PLOS Medicine

plosmedicine.org

Comments from Reviewers:

Reviewer #1: The authors have made a good effort to respond to all reviewer comments. While the study has limitations - including the high number of patients in remission (although that is also a strength as it demonstrates that IFN-a is present in inactive as well as active patients) and the absence of information on the relationship of IFN-a to clinical features of SLE (such as organ involvement), it provides a nice dataset for consideration by other investigators. It also supports the utility of the Simoa platform for detection of IFN-a. The manuscript strongly makes the case for an association of autoantibodies targeting RNA-associated proteins with serum IFN-a and a negative association of anti-phospholipid antibodies with IFN-a, with the latter an association (negative) that cries out for more investigation.

Reviewer #2: Thanks for addressing my comments and provide clarification to the methods and interpretation. One thing that I do not agree on is the use of 'data-driven' in selecting confounders. Variables that influence the exposure-outcome associations could be confounders (which should be adjusted) as well as mediators and colliders (should not be adjusted). This method does not tell us anyone on these. The authors should at least acknowledge that the method they used to induce overadjustment and collider bias.

Schisterman EF, Cole SR, Platt RW. Overadjustment bias and unnecessary adjustment in epidemiologic studies. Epidemiology. 2009 Jul 1;20(4):488-95.

Holmberg MJ, Andersen LW. Collider bias. Jama. 2022 Apr 5;327(13):1282-3.

Reviewer #3: I have carefully read the authors' responses to the questions raised during the first submission, as well as re-reviewed the revised manuscript in its entirety. The authors have satisfactorily addressed the concerns raised and modified the manuscript accordingly.

I believe that two very minor adjustments would further improve the quality of the paper:

* In the new Figure 2: please add the sample size for each subgroup (n = X). In the legend, indicate explicitly that this is a Box-and-Whisker plot, thus showing medians, quartiles, and extremes. It would also be helpful to include on the figure the thresholds derived from the ROC curve AUC values.

* Page 13, line 360: unless I am mistaken, the authors state that "the cutoff values for remission according to DORIS and LLDAS were 10,800 and …". However, in the corresponding table the value of 11,400 is reported. Could the authors clarify this apparent discrepancy?

Requests from Editors:

GENERAL

* Please confirm that your title complies with to PLOS Medicine's style. Your title must be nondeclarative and not a question. It should begin with main concept if possible. "Effect of" should be used only if causality can be inferred, i.e., for an RCT. Please place the study design ("A randomized controlled trial," "A retrospective study," "A modelling study," etc.) in the subtitle (ie, after a colon).

* Statistical reporting: Please revise throughout the manuscript, including tables and figures.

- Please report statistical information as follows to improve clarity for the reader ""22% (95% CI [13,28]; p</=)"".

- Please separate upper and lower bounds with commas instead of hyphens as the latter can be confused with reporting of negative values.

- Please repeat statistical definitions (HR, CI etc.) for each set of parentheses.

* Thank you for agreeing to make your data available. Please ensure that the Data Availability Statement in the online submission form is updated.

* Please ensure that all abbreviations are defined at first use throughout the text (including statistical abbreviations).

* Please ensure that tables and figures, including those in supplementary files, are appropriately referenced in the main text.

* Please review your text for claims of novelty or primacy (e.g. 'for the first time' or ‘novel’) and remove this language.

* Please confirm that any use of statistical terms (such as trend or significant) are supported by the data, and if not please remove them. The term trend should be used only when the test for trend has been conducted.

* Please define all acronyms used in each figure or table in its corresponding legend.

* Please revise for use of patient-centered language. Please note that patient-centered language is constructed with the use of post-modified nouns (e.g. 'patients with psoriasis’ (or similar) instead of ‘psoriasis patients’) putting the person first in the sentence structure.

* Your study is observational and therefore causality cannot be inferred. Please confirm that you have not used language that implies causality, but rather referred to associations.

* Please upload the study protocol as a "Supporting Information" file and upload an English version.

* Please confirm that the metadata in the online submission form is updated and accurate.

ABSTRACT

* Please confirm that your abstract complies with our requirements, including providing all the information relevant to this study type https://journals.plos.org/plosmedicine/s/submission-guidelines#loc-abstract

* Please confirm that all numbers presented in the abstract are present and identical to numbers presented in the main manuscript text.

* In the abstract, please include the important dependent variables that are adjusted for in the analyses.

* Please include the study design, population (with baseline characteristics) and setting, number of participants, years during which the study took place, length of follow up, and main outcome measures.

* Please quantify the main results (with 95% CIs and p values).

* l.62, “demographic characteristics and traditional cardiovascular risk factors” – please specify.

* Abstract Conclusions:

- Please address the study implications without overreaching what can be concluded from the data; the phrase "In this study, we observed ..." may be useful.

- Please interpret the study based on the results presented in the abstract, emphasizing what is new without overstating your conclusions.

METHODS AND RESULTS

* Please confirm that the ethical approval number provided covers ethical approval from all four participating hospitals.

* Thank you for providing your STROBE checklist. Please replace the page numbers with paragraph numbers per section (e.g. "Methods, paragraph 1"), since the page numbers of the final published paper may be different from the page numbers in the current manuscript.

* Table 1/2: Please change ‘Female, n (%)’ to ‘Sex, female, n (%)’.

* l.268ff, “While most SLE patients had no activity (45%), mild or moderate-high activity was present in respectively 46% and 9% of the patients as indicated by the SLEDAI score.” – It seems that most patients had mild or moderate-high activity.

* l.272, “The SDI was 0 (IQR 0-1) and an SDI score of 1 or higher was found in 46% of patients.” – We assume this refers to "SLICC-DI" in Table 1. We recommend using the same abbreviation consistently throughout.

* Figure 1: Please note that in the graph the p-value is 0.000. Also, would it be useful to align the y axis for the two distribution graphs to facilitate comparison?

* l.292ff: Please clarify that you are discussing the results of the multivariable adjustment.

* l.293ff: Please ensure to present the results with sufficient detail, e.g. SLEDAI-2K category ‘mild’ was not significantly associated with elevated IFN-α levels.

* S1 Table: For SLEDAS, it says ‘2.08’ while in the main text you write ‘2.8’ (l.316). Please check.

* l.321, “while levels above 29,000 femtograms/ml discriminated high disease activity” – Should the reader be able to find this number in Table S1? Please carefully check the numbers reported in the paragraph (vs. the table).

* l.322, “The cutoff values for remission according to DORIS and LLDAS were 10,800…” – according to S1 Table the cutoff value for DORIS is 11,400.

* Figure 1: Please include the n-numbers for each graph. Are the underlying numbers available in the Zenodo files?

* Figure 2: Please define all elements of box plots in the figure caption - center line, box limits and whiskers. Please include the n-number. Please specify ‘no activity’ (‘no disease activity’). Given a broad medical audience, we recommend that you provide a more detailed definition of "DORIS remission" and "LLDAS remission."

* Please provide the unadjusted comparisons as well as the adjusted comparisons in all relevant Tables (or as supplementary analysis).

* Please confirm that you have specified the variables controlled for in all relevant Tables.

DISCUSSION

* When revising the discussion, please aim to streamline it and keep in mind that the journal is intended for a broad medical audience.

* l.361, “In this regard, our study represents the largest investigation to date…” – please add, ‘to our knowledge’.

* l.459, “In conclusion, elevated serum levels of IFN-α identify patients with SLE exhibiting higher

460 disease activity, alongside increased ANA expression.” – please revise to focus on the associational aspect of the study.

General Editorial Requests

---

## [Editor Report · Decision Letter 3]

18 Nov 2025

Dear Dr Ferraz-Amaro, 

On behalf of my colleagues and the Guest Academic Editor, Mary K Crow, I am pleased to inform you that we have agreed to publish your manuscript "Ultrasensitive Quantification of Serum IFN-α and IFN-γ in Systemic Lupus Erythematosus: A Cross-Sectional Observational Study" (PMEDICINE-D-25-02814R3) in PLOS Medicine.

I appreciate your thorough responses to the reviewers' and editors' comments throughout the editorial process.

PRESS

Sincerely, 

Alexandra Tosun, PhD 

Senior Editor 

PLOS Medicine